# Synthesis and Characterization of 1,10-Phenanthroline-mono-*N*-oxides

**DOI:** 10.3390/molecules26123632

**Published:** 2021-06-14

**Authors:** Ferenc Najóczki, Mária Szabó, Norbert Lihi, Antal Udvardy, István Fábián

**Affiliations:** 1Department of Inorganic and Analytical Chemistry, University of Debrecen, Egyetem tér 1, H-4032 Debrecen, Hungary; najoczki.ferenc@science.unideb.hu (F.N.); szabo.maria@science.unideb.hu (M.S.); lihi.norbert@science.unideb.hu (N.L.); 2Doctoral School of Chemistry, University of Debrecen, Egyetem tér 1, H-4032 Debrecen, Hungary; 3MTA-DE Homogeneous Catalysis and Reaction Mechanisms Research Group, University of Debrecen, Egyetem tér 1, H-4032 Debrecen, Hungary; 4Department of Physical Chemistry, University of Debrecen, Egyetem tér 1, H-4032 Debrecen, Hungary; udvardya@unideb.hu

**Keywords:** density functional calculation, N-oxide, oxidation, peroxomonosulfate ion, X-ray diffraction

## Abstract

*N*-oxides of *N*-heteroaromatic compounds find widespread applications in various fields of chemistry. Although the strictly planar aromatic structure of 1,10-phenanthroline (phen) is expected to induce unique features of the corresponding *N*-oxides, so far the potential of these compounds has not been explored. In fact, appropriate procedure has not been reported for synthesizing these derivatives of phen. Now, we provide a straightforward method for the synthesis of a series of mono-*N*-oxides of 1,10-phenanthrolines. The parent compounds were oxidized by a green oxidant, peroxomonosulfate ion in acidic aqueous solution. The products were obtained in high quality and at good to excellent yields. A systematic study reveals a clear-cut correlation between the basicity of the compounds and the electronic effects of the substituents on the aromatic ring. The UV spectra of these compounds were predicted by DFT calculations at the TD-DFT/TPSSh/*def2*-TZVP level of theory.

## 1. Introduction

*N*-oxides of heterocycles are of great importance due to their widespread applicability, including, but not limited to, manufacturing chirality chemosensors [1], oxidizing agents in annulation reactions [2], intramolecular oxidants in Baeyer–Villiger reaction of ketones [3], directing group and source of oxygen atom in sulfonylation reactions [4], phase-transfer catalysts in enantioselective transformations [5], starting materials in *C-C* bond forming processes [6,7,8,9], or in the synthesis of 2-aminopyridines [10] and *N*-azine sulfoximines [11]. Recently published papers have discussed the cycloaddition reactions of *N*-oxides [12], their photochemistry [13,14] and their applications in organocatalysis [15,16,17]. Within this family of compounds, only limited information is available on 1,10-phenanthroline-1-*N*-oxide (phenO) and its derivatives. A few metal complexes of phenO have been synthesized in solid phase since the 1970s, but the detailed characterization of these compounds is not available [18,19,20,21,22,23,24,25,26,27,28]. Recent results show that platinum(II) complexes of phenO have broad potential as oxygen atom transfer reagents because of their ability to activate the N–O bond [29].

The *N*-oxidation of 1,10-phenanthroline (phen) has been studied since the 1940s [30]. It was shown that the use of H_2_O_2_ or its derivatives such as H_2_O_2_ in glacial acetic acid (where the active oxidant is most likely peracetic acid) [31,32,33] or the adduct between H_2_O_2_ and urea (known as UHP) [34] results in phenO. Earlier, an unsuccessful effort was made to obtain phenO_2_ by refluxing phen for 2 days in 90% hydrogen peroxide, glacial acetic acid, and concentrated sulfuric acid, where Caro’s acid (peroxomonosulfuric acid, H_2_SO_5_) is the dominant form of the oxidant [33]. A plausible reason behind the fruitless synthesis under such conditions was the use of inappropriate pH. Up to now, two successful methods have been reported in the literature that yield the di-*N*-oxide of phen (phenO_2_): one applies elemental fluorine as an oxidant [35], while the other one uses peroxomonosulfate ion (PMS) in aqueous solution under neutral conditions [36].

Recently, we have shown that the pH plays an important role in the kinetics of the oxidation of phen and its derivatives by PMS [36,37]. In acidic solutions, the relatively slow oxidation reaction follows net second-order kinetics with 1:1 stoichiometry of the reactants, and the reaction yields only mono-*N*-oxide. The sluggishness of the reaction is due to the protonation of the substrate which hinders the oxidative attack on the nitrogen atoms. Under such conditions, the product is also present in protonated form (HphenO^+^) featuring a strong intramolecular hydrogen bond. This prevents the oxidation of the second nitrogen atom, i.e., the formation of the corresponding di-*N*-oxide. This pH-dependence offers a simple way for the synthesis of mono-*N*-oxides of phen derivatives under relatively mild conditions and without the interference of the formation of the di-*N*-oxides.

Apart from phenO, only the preparation of 2,9-dimethyl-1,10-phenanthroline-1-*N*-oxide (DMPO) was reported earlier in the literature [38]. Now, we report the synthesis and full characterization of a series of *N*-oxides obtained from phen derivatives (Figure 1, Table 1). The rigid *N*-heteroaromatic ring structure is a specific feature of these compounds which may lead the way for unique applications in the fields mentioned above. The main objective of this study is to prepare the corresponding *N*-oxides for further studies to explore these possibilities.

## 2. Results and Discussion

### 2.1. Synthesis of the N-oxides

On the basis of our earlier kinetic results [36,37], we developed a synthetic method to obtain solely the mono-*N*-oxides of 1,10-phenanthroline derivatives by avoiding the further oxidation of the primary products. We have shown earlier that the rate of the oxidation increases significantly by increasing the pH, and various oxidation products are formed in the excess of Oxone at neutral pH. Furthermore, phen is not converted solely into phenO, even when one equivalent of PMS is used under neutral conditions [36]. The spent reaction mixture contains approximately 10–15% of phen, 70–75% of phenO and 10–15% of phenO_2_. Thus, the most important prerequisite of the selective synthesis of phenO is the use of appropriate pH, i.e., as long as the mono-*N*-oxide is in the protonated form, the di-*N*-oxidation does not take place. 

Oxone itself is acidic due to its KHSO_4_ content, but ~0.015 M H_2_SO_4_ was also added to the reaction mixtures to provide required acidic conditions for controlled oxidation. Oxone was used in less than 10% excess over the substrate. While increased concentration of the oxidant would have reduced the reaction time significantly, such conditions were not used in order to avoid complications during further steps of the preparative process. A systematic study of the temperature dependence of the oxidation rates revealed that the reactions proceed more than an order of magnitude faster above 60 °C than at room temperature. However, it was reported earlier that temperatures above 70 °C may result in the opening of the middle ring of the substrate [38]. Thus, the oxidation was performed at 60 °C by strictly controlling the temperature. The progress of the reaction was monitored by analyzing the reaction mixture for the substrate and the product by the HPLC method. In the HPLC chromatograms of the spent reaction mixtures, the characteristic peak of the substrate was absent, and only one intense new peak appeared, which was assigned to the product (Appendix A). The quantitative (more than 99 %) conversion of the substrates into the corresponding mono-*N*-oxides was also confirmed by conventional 1D ^1^H and ^13^C NMR measurements, 2D correlation (^1^H-^1^H and ^1^H-^13^C) techniques, and mass spectrometry (Appendix A). The chemical shifts are listed in the Appendix A. The results are in excellent agreement with the data reported for phenO [29] and DMPO [38].

The time required for complete consumption of the substrate varied between 2 and 38 h (Table 2) depending on the substituents on phen. In order to obtain solid products, an aliquot of 2 M NaOH solution was added to the reaction mixture to set the pH between 9 and 10. Due to the pH change, the color of the solution turned into deep orange. Subsequently, the reaction mixture was extracted with appropriate amount of chloroform. Finally, the organic solvent was removed in a vacuum rotational evaporator. The color of the final products varied between yellow and purple depending on the substituent. The hydration of these dry materials led to a brownish color change.

The desired products were obtained from good to excellent yields (Table 2). When isomer *N*-oxides are formed, the yield of the product is given before separation of the isomers. As indicated in our previous paper [37], the 5MPO-6MPO, 5NPO-6NPO and 5CPO-6CPO isomer pairs form in about a 1:1 ratio, while 3 times more 4MPO is produced than 7MPO in the oxidation of 4MP. Finally, the isomers were separated by preparative HPLC method.

### 2.2. The Acid Dissociation Constants of Substituted Mono-N-oxide Derivatives

Proper characterization of the acid-base properties of the synthetized *N*-heteroaromatic compounds are of great significance for various reasons. These features have crucial implications in the coordination and redox chemistry of these compounds, as well as in their practical applications. Furthermore, the p*K*_a_-s of these compounds are also important regarding the synthesis of mono-*N*-oxides, because they need to be in protonated form during the oxidation of the parent compounds in order to avoid the formation of further oxidation products. In contrast, the sufficient extraction of the products from aqueous solution by organic solvents requires that they are present mainly in deprotonated (neutral) form.

The acid dissociation constants of the phenanthrolinium ion (Equation (1)) of phenO-s were determined by one of the methods discussed in the subsequent paragraphs.
(1)$${\mathrm{HL}}^{+}\;⇌\;L+H^{+}\;K_{a}=\frac{\left[L\right]\left[H^{+}\right]}{\left[{\mathrm{LH}}^{+}\right]}$$
where L denotes the *N*-oxide.

When the solubility of the substrate was sufficiently high, standard pH-potentiometric titration was performed as described earlier [39]. The relatively small solubility of TMPO and DMPO at 25.0 °C prevented the use of pH-potentiometric titration, and a combined pH-potentiometric and spectrophotometric method was used, where the UV-Vis spectra of the samples were recorded as a function of pH (Figure 2) [40]. 

The compounds reported here typically feature intense absorption bands in the UV region with *ε* = 20,000–30,000 M^−1^cm^−1^. Thus, reliable spectrophotometric measurements are feasible in their dilute solutions at concentration levels as low as ~10–100 μM. Acidified solutions of L (ionic strength is set to the desired value) were titrated with standardized NaOH solution and the pH and the spectra were recorded. The absorbance data at 4–5 wavelengths were fitted simultaneously to Equation (2) (Figure 2).
(2)$$A=\frac{\varepsilon_{\mathrm{HL}}\left[H^{+}\right]+\varepsilon_{L}K_{a}}{\left[H^{+}\right]+K_{a}}c_{\mathrm{tot}}$$
where *A* is the absorbance measured at a given [H]^+^ and wavelength (*λ*), *ε*_HL_ and *ε*_L_ are the molar absorption coefficients for the protonated and deprotonated forms at *λ*, *K*_a_ is the acid dissociation constant and *c*_tot_ is the analytical concentration of the substrate.

In several cases, the *N*-oxidation of the substrate yields structural isomers of very similar basicity. The pH-potentiometric and spectrophotometric methods are not suitable to simultaneously determine the corresponding p*K*_a_-s without separation of these compounds because the protolytic equilibria strongly overlap. In such a situation, ^1^H NMR spectra were recorded in the solutions of the unseparated isomers. The chemical shifts of the NMR peaks of each isomer were selectively followed as a function of pH and used for calculating the p*K*_a_-s. As an example, the pH dependent ^1^H NMR spectra of the 4MPO-7MPO system (the isomer oxidation products of 4MP) are shown in Figure 3A. Each isomer has 7 chemically inequivalent aromatic and 3 equivalent methyl protons. The assignment of the aromatic proton peaks corresponds to Figure 1. (pH-dependent ^1^H NMR spectra of the 5CPO-6CPO, 5NPO-6NPO, 5MPO-6MPO pairs are reported in Appendix A.). The goodness of the fit is demonstrated in Figure 3B, where the relative chemical shift (Equation (3)) is plotted as a function of pH.
(3)$$\delta_{\mathrm{rel}}=\frac{\delta_{\mathrm{pH}}–\delta_{L}}{\delta_{\mathrm{HL}}–\delta_{L}}\;$$

Some of the peaks overlap at a specific pH and cannot be used for calculating the p*K*_a_-s. Finally, the chemical shifts of three 4MPO (methyl, A2, A9) peaks, as well as three 7MPO (methyl, B2, B9) peaks, were fitted simultaneously to Equation (4).
(4)$$\delta =\frac{\delta_{\mathrm{HL}}{\left[H\right]}^{+}+\delta_{L}K_{a}}{\left[H^{+}\right]+K_{a}}\;$$
where *δ* is the chemical shift measured at a given [H^+^], *δ*_HL_ and *δ*_L_ are the chemical shifts for the protonated and deprotonated forms. 

The identical pH patterns of the chemical shifts of the peaks of the given isomer also corroborate the peak assignments of the ^1^H NMR spectra.

The calculated p*K*_a_-s are listed in Table 3. Basicity data for phenO-s have not been reported before.

The comparison of the data in Table 3 reveals that phen and its derivatives are significantly (typically about 1.5 to 2.5 p*K*_a_ units) stronger acids than the corresponding mono-*N*-oxides. In our recent study, we reported unequivocal evidence for the formation of a strong intramolecular hydrogen bond in HphenO^+^ where the NH^+^ proton partially bonds to the NO oxygen and a six-membered ring forms [36]. This structure stabilizes the protonated species and reduces the driving force of the acid dissociation. 

The interpretation of the trends in the p*K*_a_-s for both sets of the 5-substituted compounds (phen-s and phenO-s) is straightforward when the electronic effects of the substituents are considered. Electron withdrawing groups (EWG); such as -NO_2_, -Cl, reduce the electron density of the aromatic rings and make the molecule more acidic, while an electron donating group (EDG; methyl in our case) decreases the acid strength compared to the non-substituted molecules (phen and phenO).

The distance of the substituent from the protonation site is also a key factor in these compounds. In the *N*-oxides, the protonation center is the unoxidized nitrogen atom. The closer of the substituent is to the unoxidized nitrogen atom the bigger its electronic contribution to the p*K*_a_. Pairwise comparison of the acidities of the corresponding derivatives clearly demonstrates this effect: p*K*_a_(6CPO) < p*K*_a_(5CPO); p*K*_a_(6NPO) < p*K*_a_(5NPO); and p*K*_a_(4MPO) < p*K*_a_(5MPO) < p*K*_a_(6MPO) < p*K*_a_(7MPO).

Finally, the same reasoning is valid when the acidities of the mono-, di- and tetramethyl derivatives are compared. The p*K*_a_ increases in both series when the number of the electron donating methyl groups is increased.
p*K*_a_(phen) < p*K*_a_(4MP)/p*K*_a_(5MP) < p*K*_a_(DMP) < p*K*_a_(TMP)
and p*K*_a_(phenO) < p*K*_a_(4MPO)/p*K*_a_(5MPO) < p*K*_a_(DMPO) < p*K*_a_(TMPO)

### 2.3. X-ray Structures of 1,10-Phenanthroline-1-N-oxide Derivatives

The structures of three new 1,10-phenanthroline-mono-*N*-oxide derivatives were analyzed by X-ray diffractometry. Good quality crystals of 5CPO×H_2_O and 6CPO×H_2_O were obtained from their concentrated non-anhydrous methanolic solutions. Both derivatives crystallize in the monoclinic *P*2_1_/*c* space group. The corresponding structures are shown in Appendix A and the most important bond lengths are listed in Table 4 (further crystallographic data are listed in Appendix A).

The N–O bond distances in 5CPO×H_2_O and 6CPO×H_2_O (1.294(4) Å, 1.306(2) Å, respectively) compare well with the average of N–O distances observed in other 1,10-phenanthroline-mono-*N*-oxides. [38,41] The chemical occupancies of the oxygen atoms 1 (100%) are consistent with the formation of mono-*N*-oxides. In both cases, the 15 atoms of the ring system are essentially located in one plane, so the *N*-oxidation does not lead to the distortion of the planar aromatic structure of the 1,10-phenanthroline backbone. The asymmetric units of the two compounds contain one water molecule as a solvent which links the polar fragments (Appendix A). The organic molecules are layered at a distance of 3.494 Å and 3.392 Å, respectively. The parallel packing structures of these compounds are stabilized by the stacking interactions between the aromatic rings and the hydrogen bond network (Appendix A), as shown in Figure 4.

The 6CPO isomer was also crystallized from chloroform by slow evaporation. In this case, crystals of the orthorhombic *Pbca* space group were obtained and the asymmetric unit contains only the mono-*N*-oxide (Appendix A). As expected, the molecular structures of 6CPO and 6CPO×H_2_O are very similar, although the N–O distance is somewhat shorter in 6CPO, 2.284(3) Å.

In the case of 5CPO, 6CPO and 6CPOw, the calculated distortion angle between N1, C10B, C10A and N10 (3.3(5)°, 0.9(3)° and 0.5(3)°, respectively) practically indicates planar geometry.

TMPO was crystallized from non-anhydrous methanolic solution but the diffraction pattern of the crystals is weak. A number of data sets were also collected but in the case of the best sets the *R*_1_ and w*R*_2_ are 9.81% and 30.92%. While this prevented the determination of the exact bond lengths, the connectivities of TMPO are clearly established and are in good agreement with the results of the ^1^H-, ^13^C-NMR data (Appendix A).

TMPO crystallizes in the triclinic crystal system $\bar{P}1$ space group (as 2,9-dimethyl-1,10-phenanthroline-1-*N*-oxide dihydrate [38]), the asymmetric unit contains two mono-*N*-oxides linked by hydrogen network of four water molecules as shown in Figure 5 and Appendix A.

### 2.4. DFT Calculations

In order to elucidate the electronic absorption spectra and the nature of the electronic transitions of the *N*-oxides, DFT and TD-DFT calculations were performed (Cartesian coordinates are reported in Appendix A). The comparison of all calculated data (using different functionals) with the experimental spectra revealed that the 10% exact exchange as represented by the TPSSh functional combined with the triple-ζ *def2*-TZVP basis set provides the best prediction of the UV-Vis spectra of these compounds. The experimental and calculated spectra are compared for phenO in Figure 6 (spectra for the other phenO-s are shown in Appendix A).

All spectra exhibit three or four characteristic absorption bands in the 200–400 nm wavelength range (Appendix A). The most intensive band at 260–280 nm features the superposition of two transitions, which are due to a combination of excitations from occupied MOs with π_arom_ character. The absorptions are mainly associated with the π-π* transitions of the aromatic system and the oscillator strengths are consistent with the experimental values of molar absorptivities. It is also noticeable, that neither the electron donating nor the electron withdrawing groups have significant effects on the energy of the main transition. This also confirms that the excitation is associated with the MOs bearing aromatic character.

The ^1^H NMR chemical shifts of phenO were calculated by using the GIAO approach. The experimental and the predicted parameters are in excellent agreement (Figure 7).

## 3. Materials and Methods

### 3.1. General Information

All reagents and solvents were of analytical grade and were used as received from commercial sources. All the studied phen-s are commercially available (Aldrich) and were used without further purification. Potassium peroxomonosulfate is available as a stable triple salt, a brand named Oxone (2KHSO_5_·KHSO_4_·K_2_SO_4_, Aldrich) and the solid was added directly to the reaction mixtures during the synthesis of the *N*-oxides.

UV-Vis spectra were recorded on scanning spectrophotometers (Shimadzu UV-1800, Shimadzu, Duisburg, Germany) at a constant temperature maintained by the use of thermostats attached to the instruments. All measurements were performed at 25.0 ± 0.1 °C. Standard 1.000 cm quartz cuvettes were used.

The pH-metric and iodometric measurements were performed with a *Metrohm 785 DMP Titrino* (Metrohm Magyarország, Budapest, Hungary) automatic titrator equipped with 6.0262.100 and 6.0451.100 combined electrodes, respectively. The pH-electrode was calibrated by two buffers according to IUPAC recommendations [42]. The pH readings were converted to hydrogen ion concentration, as described by Irving et al. [43]. The acid-base equilibria of the *N*-oxides were studied by various methods: standard pH-potentiometric titration, combined pH-potentiometric and UV-vis spectrophotometric method or combined pH-potentiometric and ^1^H NMR technique.

ESI-TOF-MS measurements were made with a Bruker maXis II MicroTOF-Q type Qq-TOF-MS instrument (Bruker Daltonik, Bremen, Germany) in positive mode. The instrument was equipped with an electrospray ion source where the spray voltage was 4 kV. N_2_ was utilized as a drying gas and the drying temperature was 200 °C. The spectra were accumulated and recorded using a digitalizer at a sampling rate of 2 GHz. The mass spectra were calibrated externally using the exact masses of the clusters generated from the electrosprayed solution of sodium trifluoroacetate (NaTFA). The spectra were analyzed with the *DataAnalysis 4.4* software from Bruker.

1D ^1^H, ^13^C and 2D (^1^H–^1^H COSY, and ^1^H–^13^C HSQC, HMBC) NMR spectra were recorded on a Bruker Avance I 400 spectrometer ((Bruker Daltonik, Bremen, Germany) equipped a BB inverse *z* gradient probe at 298 K in CDCl_3_ solution with TMS, as an internal standard (*δ* = 0 ppm). The COSY spectrum of phenO was obtained at 400 Hz in D_2_O solution with TMS, too. In the combined pH-potentiometric and ^1^H NMR method, the spectra were measured at 400 MHz. The solutions were prepared in H_2_O, and DSS (4,4-dimethyl-4-silapentane-1-sulfonic acid) in D_2_O was added to the sample in a capillary as an external standard for ^1^H (0 ppm). The ^1^H NMR spectra were recorded by using the standard watergate pulse sequence for the suppression of water proton signal. In each experiment, 32 scans were collected with 16K data points using a sweep width of 5995 Hz, a pulse angle of 90°, an acquisition time of 1.366 s, and a relaxation delay of 1 s. The HSQC spectra were collected by using gradient pulses in the *z* direction with the standard Bruker pulse sequence.

The isomer *N*-oxides were separated by a *YL 9100* preparative HPLC system equipped with a *YL 9120S UV/Vis* detector and using *Phenomenex Luna C18* column Nr 429545-1.

Non-linear least squares fittings of the different types of titration curves were performed with the software *Scientist* [44].

The ionic strength was kept constant by using appropriate amounts of sodium nitrate or sodium chloride in all experiments.

### 3.2. The Preparation of 1,10-Phenanthroline-1-N-oxides

A ~15 mM aqueous solution of the organic substrate was prepared. Small amounts of sulfuric acid were added to the solution to increase the solubility of the substrate and to provide slightly acidic conditions (pH ~ 2), which prevents di-*N*-oxidation. About 1.1-1.2 equivalent of solid PMS was added and the mixture was stirred at 60 °C for 2–38 h. After complete conversion, the reaction mixture was neutralized by adding NaOH solution and the pH was set about 3–3.5 pH unit above the p*K*_a_ of the initial phen derivative to ensure that the produced *N*-oxide completely deprotonates. After repeated extraction with CHCl_3_ (10 mL × 3 times), the combined organic extract was dried over anhydrous Na_2_SO_4_, filtered, and the solvent was removed under reduced pressure. In the case of non-symmetrical initial phen derivatives, the oxidation results in the formation of structurally isomer mono-*N*-oxides. The isomers were separated by preparative HPLC. The separated fractions were concentrated to 1/10 of the initial volume, extracted by CHCl_3_, the extract was dried over anhydrous Na_2_SO_4_, filtered, and the solvent was removed under reduced pressure.

*1,10-phenanthroline-1-N-oxide (phenO).* 1,10-Phenanthroline (phen) (150.26 mg, 0.758 mmol) was dissolved in water under acidic conditions (30 mL H_2_O, 200 μL 2 M H_2_SO_4_, pH < 2) and was reacted with potassium peroxomonosulfate (PMS) (276.95 mg, 0.778 mmol). The general procedure was followed and yellowish-brown solid was obtained (128.91 mg, 0.65 mmol, yield 86.6%). ^1^H NMR (400 MHz, CDCl_3_, 25 °C, TMS) *δ* (ppm): 7.35 (dd, *J* = 6.5, 1.4 Hz, 1H, ***8***), 7.55 (dd, *J* = 8.0, 4.5 Hz, 1H, ***3***), 7.62 (m, 1H, ***7***), 7.66 (m, 1H, ***5***), 7.68 (m, 1H, ***6***), 8.11 (dd, *J* = 8.0, 1.4 Hz 1H, ***4***), 8.62 (d, *J* = 6.5 Hz, 1H, ***9***), 9.19 (dd, *J* = 4.5, 1.5 Hz, 1H, ***2***). ^13^C NMR (100.6 MHz, CDCl_3_, 25°C, TMS) *δ* (ppm): 121.8 (C8), 122.1 (C3), 123.8 (C7), 125.4 (C6), 127.9 (C4A), 128.0 (C5), 132.2 (C6A), 134.9(C4), 137.1 (C10A), 139.6 (C9), 141.4 (C10B), 148.8 (C2). ESI-MS calculated (calcd.) for C_12_H_9_N_2_O^+^ [M+H]^+^ = 197.0709, found 197.0712.

*2,9-dimethyl-1,10-phenanthroline-1-N-oxide (DMPO).* 2,9-Dimethyl-1,10-phenanthroline (DMP) (100.7 mg, 0.484 mmol) was dissolved in water under acidic conditions (30 mL H_2_O, 100 μL 2 M H_2_SO_4_, pH < 2) and was reacted with PMS (177.0 mg, 0.500 mmol). The general procedure was followed, and dark purple solid was obtained (91.0 mg, 0.406 mmol, yield 83.91%). ^1^H NMR (400 MHz, CDCl_3_, 25 °C, TMS) *δ* (ppm): 2.69 (s, 3H, ***2***), 2.86 (s, 3H, ***9***), 7.42 (d, *J* = 3.0 Hz, 1H, ***7***), 7.43 (d, *J* = 3.0 Hz, 1H, ***4***), 7.54 (d, *J* = 7.7 Hz, 1H, ***8***), 7.58 (d, *J* = 8.6 Hz, 1H, ***6***), 7.62 (d, *J* = 8.6 Hz, 1H, ***5***), 8.02 (d, *J* = 7.6 Hz, 1H, ***3***). ^13^C NMR (100.6 MHz, CDCl_3_, 25 °C, TMS) *δ* (ppm): 18.5 (–**C**H_3_(9)), 25.0 (–**C**H_3_(2)), 122.4 (C7), 122.6 (C4), 123.0 (C8), 124.5 (C6), 126.0 (C10A), 126.9 (C5), 130.8 (C6A), 134.9 (C3), 137.0 (C4A), 141.2 (C10B), 149.2 (C9A), 157.7 (C2A). ESI-MS calcd. for C_14_H_13_N_2_O^+^ [M+H]^+^ = 225.1022, found 225.1025, calcd. for C_14_H_12_N_2_ONa^+^ [M+Na]^+^ = 247.0842, found 247.0840.

*3,4,7,8-tetramethyl-1,10-phenanthroline-1-N-oxide (TMPO).* 3,4,7,8-Tetramethyl-1,10-phenanthroline (TMP) (150.12 mg, 0.635 mmol) was dissolved in water under acidic conditions (35 mL H_2_O, 200 μL 2 M H_2_SO_4_, pH < 2) and was reacted with PMS (226.91 mg, 0.637 mmol). The general procedure was followed, and yellow solid was obtained (141.26 mg, 0.559 mmol, yield 88.1%). ^1^H NMR (400 MHz, CDCl_3_, 25 °C, TMS) *δ* (ppm): 2.43 (s, 3H, ***3***), 2.53 (s, 3H, ***4***), 2.61 (s, 3H, ***8***), 2.67 (s, 3H, ***7***), 7.94 (d, *J* = 9.6 Hz, 1H, ***6***), 8.06 (d, *J* = 9.6 Hz, 1H, ***5***), 8.60 (s, 1H, ***9***), 9.02 (s, 1H, ***2***). ^13^C NMR (100.6 MHz, CDCl_3_, 25 °C, TMS) *δ* (ppm): 14.8 (C11), 15.0 (C12), 17.7(C13, C14), 122.1 (C6), 124.5 (C5), 127.3 (C3), 130.9 (C6A), 131.2 (C4,C7), 131.6 (C4A), 136.8 (C8), 141.0 (C10A), 141.1 (C9), 141.4 (C10B), 151.0 (C2). ESI-MS calcd. for C_16_H_17_N_2_O^+^ [M+H]^+^ = 253.1335, found 253.1335.

*4-methyl-1,10-phenanthroline-1-N-oxide (4MPO) and 7-methyl-1,10-phenanthroline-1-N-oxide (7MPO).* 4-Methyl-1,10-phenanthroline (4MP) (150.12 mg, 0.776 mmol) was dissolved in water under acidic conditions (30 mL H_2_O, 100 μL 2 M H_2_SO_4_, pH < 2) and was reacted with PMS (275.38 mg, 0.773 mmol). The general procedure was followed and dark green solid was obtained (134.65 mg, 0.643 mmol, overall yield for the two isomers 82.8 %). The separation of the two isomers was achieved on Luna® Prep C18(2) (100 Å, 250 mm × 21.2 mm; 5 μm) column using isocratic elution with 16 % Na_2_HPO_4_ (5 mM in water): 76% NaH_2_PO_4_ (5 mM in water): 8% ACN solvent mixture as a mobile phase using 25 mL/min flow-rate.

*4-methyl-1,10-phenanthroline-1-N-oxide (4MPO):* HPLC t_R_ = 11.0 min. ^1^H NMR (400 MHz, D_2_O, 25 °C, TMS) *δ* (ppm): 3.01 (s, 3H, –C***H_3_***), 8.01 (dd, *J* = 8.3, 6.4 Hz, 1H, ***8***), 8.15 (d, 1H, *J* = 9.3 Hz, ***5***), 8.20 (d, *J* = 8.2 Hz, 1H, ***3***), 8.26 (d, *J* = 9.3 Hz, 1H, ***6***), 8.38 (d, *J* = 8.4 Hz, 1H, ***7***), 8.87 (dd, *J* = 6.3, 0.7 Hz, 1H, ***9***), 9.00 (d, *J* = 5.8 Hz, 1H, ***9***). ^13^C NMR (100.6 MHz, D_2_O, 25 °C, TMS) *δ* (ppm): 19.1 (–**C**H_3_), 123.9 (C6), 125.7 (C8), 126.0 (C3), 127.9 (C6A), 128.3 (C5), 128.5 (C4), 131.2 (C4), 131.4 (C7), 132.1 (C10A), 139.6 (C9), 139.8 (C2), 157.1 (C10B). ESI-MS calcd. for C_13_H_11_N_2_O^+^ [M+H]^+^ = 211.0866, found 211.0864, calcd. for C_13_H_11_N_2_ONa^+^ [M+Na]^+^ = 233.0685 found 233.0681.

*7-methyl-1,10-phenanthroline-1-N-oxide (7MPO):* HPLC t_R_ = 15.9 min. ^1^H NMR (400 MHz, D_2_O, 25 °C, TMS) *δ* (ppm): 2.83 (s, 3H, –C***H_3_***), 7.86 (d, *J* = 6.5 Hz, 1H, ***8***), 8.25 (d, 1H, *J* = 9.3 Hz, ***5***), 8.30 (d, *J* = 9.3 Hz, 1H, ***6***), 8.37 (dd, *J* = 8.4, 5.5 Hz, 1H, ***3***), 8.78 (d, *J* = 6.5 Hz, 1H, ***9***), 9.16 (dd, *J* = 8.5, 1.3 Hz, 1H, ***4***), 9.22 (dd, *J* = 5.6, 1.4 Hz, 1H, ***2***). ^13^C NMR (100.6 MHz, D_2_O, 25°C, TMS) *δ* (ppm): 17.7 (–**C**H_3_), 125.1 (C3), 125.3 (C6), 126.1 (C8), 127.1 (C5), 127.4 (C6A), 129.2 (C4A), 131.3 (C7), 132.7 (C10A), 138.4 (C9), 141.2 (C2), 142.2 (C10B), 144.4 (C4). ESI-MS calcd. for C_13_H_11_N_2_O^+^ [M+H]^+^ = 211.0866, found 211.0865, calcd. for C_13_H_11_N_2_ONa^+^ [M+Na]^+^ = 233.0685 found 233.0682.

*5-methyl-1,10-phenanthroline-1-N-oxide (5MPO) and 6-methyl-1,10-phenanthroline-N-oxide (6MPO).*5-Methyl-1,10-phenanthroline (5MP) (150.27 mg, 0.777 mmol) was dissolved in water under acidic conditions (35 mL H_2_O, 200 μL 2 M H_2_SO_4_, pH < 2) and was reacted with PMS (279.60 mg, 0.785 mmol). The general procedure was followed, and pale brown solid was obtained (129.36 mg, 0.618 mmol, overall yield for the two isomers 79.5%). The preparative separation of the two isomers was achieved on Luna® Prep C18(2) (100 Å, 250 mm × 21.2 mm; 5 μm) column using isocratic elution with 35 *v/v*% Na_2_HPO_4_ (5 mM in water): 55 *v/v*% NaH_2_PO_4_ (5 mM in water):10 *v/v*% ACN solvent mixture as a mobile phase using 25 mL/min flow-rate.

*5-methyl-1,10-fenantroline-1-N-oxide (5MPO):* HPLC: t_R_ = 21.8 min. ^1^H NMR (400 MHz, D_2_O, 25 °C, TMS) δ (ppm): 2.48 (s, 3H, –C***H_3_***), 7.55 (s, 1H, ***6***), 7.83 (dd, *J* = 8.6, 6.3 Hz, 1H, ***8***), 8.08 (dd, *J* = 8.3, 5.2 Hz, 1H, ***3***), 8.18 (dd, *J* = 8.6, 0.6 Hz, 1H, ***7***), 8.61 (dd, *J* = 8.3, 1.3 Hz, 1H, ***4***), 8.68 (dd, *J* = 6.3, 0.6 Hz, 1H, ***9***), 8.95 (dd, J = 5.3, 1.4 Hz, 1H, ***2***). ^13^C NMR (100.6 MHz, D_2_O, 25°C, TMS) δ (ppm): 17.7 (–***C***H_3_), 124.2 (C8), 124.4 (C3), 125.8 (C6), 126.8 (C7), 128.2 (C10B), 128.5 (C6A), 131.6 (C5), 131.7 (C4A), 135.4 (C10A), 138.5 (C2), 141.2 (C9), 141.5 (C4). ESI-MS calcd. for C_13_H_11_N_2_O^+^ [M+H]^+^ = 211.0866, found 211.0868.

*6-methyl-1,10-fenantroline-1-N-oxide (6MPO):* HPLC: t_R_ = 24.5 min. ^1^H NMR (400 MHz, D_2_O, 25 °C, TMS) δ (ppm): 2.32 (s, 3H, –C***H_3_***), 7.25 (s, 1H, ***5***), 7.64 (dd, *J* = 8.1, 6.3 Hz, 1H, ***8***), 7.77 (d, *J* = 8.3 Hz, 1H, ***7***), 7.95 (dd, *J* = 8.4, 5.0 Hz, 1H, ***3***), 8.49 (d, *J* = 6.2 Hz, 1H, ***9***), 8.57 (d, *J* = 8.2 Hz, 1H, ***4***), 8.87 (dd, *J* = 5.1, 0.9 Hz, 1H, ***2***). ^13^C NMR (100.6 MHz, D_2_O, 25°C, TMS) δ (ppm): 16.7 (–**C**H_3_), 123.7 (C3), 124.1 (C8), 125.5 (C5), 127.6 (C6A), 127.9 (C6), 129.0 (C7), 131.0 (C10A), 133.3 (C4A), 134.7 (C10B), 137.6 (C4), 138.0 (C9), 142.2 (C2). ESI-MS calcd. for C_13_H_11_N_2_O^+^ [M+H]^+^ = 211.0866, found 211.0866.

*5-chloro-1,10-phenanthroline-1-N-oxide (5CPO) and 6-chloro-1,10-phenanthroline-1-N-oxide (6CPO).* 5-Chloro-1,10-phenanthroline (5CP) (100.1 mg, 0.430 mmol) was dissolved in water under acidic conditions (30 mL H_2_O, 100 μL 3 M H_2_SO_4_, pH < 2) and was reacted with PMS (173 mg, 0.489 mmol). The general procedure was followed, and pale brown solid was obtained (86.5 mg, 0.375 mmol, overall yield for the two isomers 87.2%). The preparative separation of the two isomers was achieved on Luna® Prep C18(2) column (100 Å, 250 mm × 21.2 mm) using acetonitrile – 0.005 M citrate buffer (pH = 6.27) solvent mixture at 19/81 *v/v* ratio as a mobile phase with the flow-rate of 25 mL/min.

*5-chloro-1,10-fenantroline-1-N-oxide (5CPO):* HPLC: t_R_ = 11.4 min. ^1^H NMR (400 MHz, CDCl_3_, 25 °C, TMS) δ (ppm): 7.42 (t, *J* = 8.4, 6.9 Hz, 1H, ***8***), 7.60 (d, *J* = 8.4 Hz, 1H, ***7***), 7.72 (dd, *J* = 8.4, 4.4 Hz, 1H, ***3***), 7.82 (s, 1H, ***6***), 8.66 (s, 2H, ***4***,***9***), 9.29 (d, *J* = 3.1 Hz, 1H, ***2***). ^13^C NMR (100.6 MHz, CDCl_3_, 25 °C, TMS) δ (ppm): 122.5 (C8), 122.6 (C3, C7), 124.7 (C6), 125.7 (C6A), 131.3 (C5), 131.6 (C4A), 132.0 (C4), 136.3 (C10A), 140.1 (C9), 142.0 (C10B), 149.3 (C2). ESI-MS calcd. for C_12_H_8_N_2_OCl^+^ [M+H]^+^ = 231.0320, found 231.0325, calcd. for C_12_H_7_N_2_OClNa^+^ [M+Na]^+^ = 253.0139, found 253.0148.

*6-chloro-1,10-fenantroline-1-N-oxide (6CPO):* HPLC: t_R_ = 10.3 min. ^1^H NMR (400 MHz, CDCl_3_) δ (ppm): 7.51 (dd, *J* = 8.3, 6.4 Hz, 1H, ***8***), 7.62 (dd, *J* = 8.0, 4.4 Hz, 1H, **3**), 7.89 (s, 1H, ***5***), 8.10 (dd, *J* = 8.2, 1.4 Hz, 1H, ***4***), 8.17 (d, *J* = 8.4 Hz, ***7***), 8.72 (d, *J* = 6.3 Hz, 1H, ***9***), 9.24 (dd, *J* = 4.5, 1.6 Hz, 1H, ***2***). ^13^C NMR (100.6 MHz, CDCl_3_, 25°C, TMS) δ (ppm): 120.4 (C7), 122.1 (C8), 122.8 (C3), 127.1 (C4A), 127.4 (C5), 128.6 (C6A), 130.3 (C6), 134.2 (C4), 138.1 (C10A), 140.4 (C9), 140.6 (C10B), 149.1 (C2). ESI-MS calcd. for C_12_H_8_N_2_OCl^+^ [M+H]^+^ = 231.0320, found 231.0327, calcd. for C_12_H_7_N_2_OClNa^+^ [M+Na]^+^ = 253.0139, found 231.0150.

*5-nitro-1,10-phenanthroline-1-N-oxide (5NPO) and 6-nitro-1,10-phenanthroline-1-N-oxide (6NPO)*. 5-Nitro-1,10-phenanthroline (99.9 mg, 0.411 mmol) was dissolved in water under acidic conditions (30 mL H_2_O, 100 μL 3 M H_2_SO_4_, pH < 2) and was reacted with PMS (164 mg, 0.463 mmol). The general procedure was followed, and orange solid was obtained (91.2 mg, 0.378 mmol, overall yield for the two isomers 92.1%). Eluent used for the separation of the isomers was as follows: 0.005 M acetic acid (the pH was adjusted to 5.12 by adding an appropriate amount of NaOH) and ACN solvent mixture at 90/10 *v/v* ratio. The preparative separation of the two isomers was achieved on Luna® Prep C18(2) column (100 Å, 250 mm × 21.2 mm) using 25 mL/min flow rate.

*5-nitro-1,10-fenantroline-1-N-oxide (5NPO):* HPLC: t_R_ = 14.9 min. ^1^H NMR (400 MHz, CDCl_3_, 25 °C, TMS) δ (ppm): 7.54 (dd, *J* = 8.0, 6.5 Hz, 1H, ***8***), 7.77 (dd, *J* = 8.6, 4.4 Hz, 1H, ***3***), 7.84 (d, *J* = 8 Hz, 1H, ***7***), 8.45 (s, 1H, ***6***), 8.78 (dd, *J* = 6.4, 0.5 Hz, 1H, ***9***), 8.82 (dd, *J* = 8.6, 1.6 Hz, 1H, ***4***), 9.33 (dd, *J* = 4.4, 1.6 Hz, 1H, ***2***). ^13^C NMR (100.6 MHz, CDCl_3_, 25°C, TMS) δ (ppm): 119.5 (C6A), 123.2 (C8), 123.3 (C3), 123.5 (C6), 124.6 (C7), 129.3 (C4A), 130.7 (C4), 138.3 (C10B), 142.0 (C10A), 142.7 (C9), 144.9 (C5), 149.8 (C2). ESI-MS calcd. for C_12_H_8_N_3_O_3_^+^ [M+H]^+^ = 242.0561, found 242.0560, calcd. for C_12_H_7_N_3_O_3_Na^+^ [M+Na]^+^ = 264.0380, found 264.0380.

*6-nitro-1,10-fenantroline-1-N-oxide (6NPO):* HPLC: t_R_ = 13.4 min. ^1^H NMR (400 MHz, D_2_O (+NaOD), 25 °C, TMS) δ (ppm): 6.01 (s, 1H, ***5***), 7.53 (t, *J* = 7.3 Hz, 1H, ***8***), 7.57 (dd, *J* = 8.2, 4.7 Hz, 1H, ***3***), 7.74 (d, *J* = 7.7 Hz, 1H, ***7***), 8.31 (d, *J* = 6.5 Hz, 1H, ***9***), 8.58 (dd, *J* = 4.8, 1.0 Hz, 1H, ***4***), 8.81 (dd, *J* = 8.5, 1 Hz, 1H, ***2***). ^13^C NMR (100.6 MHz, D_2_O (+NaOD), 25°C, TMS) δ (ppm): 66.0 (C5), 121.2 (C6A), 123.6 (C3), 124.9 (C8), 127.6 (C4A), 129.4 (C7), 136.6 (C2), 138.9 (C10B), 139.0 (C9), 141.8 (C10A), 142.3 (C6), 145.7 (C4). ESI-MS calcd. for C_12_H_8_N_3_O_3_^+^ [M+H]^+^ = 242.0565, found 242.0560, calcd. for C_12_H_7_N_3_O_3_Na^+^ [M+Na]^+^ = 264.0389, found 264.0380.

### 3.3. X-ray Structures of 1,10-Phenatroline-1-N-oxide Derivatives

Suitable single crystals of 1,10-phenatroline-1-*N*-oxide derivatives were mounted on the Mitegen loops with oil. Data sets were collected at 100 K or room temperature on a Bruker D8 Venture (SC-XRD) diffractometer (Bruker Daltonik, Bremen, Germany) system using INCOATEC IμS 3.0 dual (Mo, Cu) sealed tube microsources (Mo-*K*α irradiation (λ = 0.71073 Å) was applied for all measurements). and Photon II Charge-integrating Pixel Array detector. Bruker APEX3 software was applied to collect and made the absorption correction using the MULTI-SCAN method and integration of the data sets [45]. The structures were solved by the direct method using SHELXT [46] and refined on *F*^2^ data using full matrix least-squares by SHELXL [47], were managed with OLEX^2^ [48] and WinGX software suites [49]. All non-hydrogen atoms were refined anisotropically. All hydrogens were included in the model at geometrically calculated positions and refined using the riding model. OH (water) hydrogens were located on the difference electron density map.

The optimized structures of the compounds were analyzed using PLATON [50]; publication materials were prepared with the Mercury CSD-4.3.0 [51] and OLEX^2^ software.

The crystallographic data for all compounds were deposited in the Cambridge Crystallographic Data Centre (CCDC) with the No. CCDC 2075043, 2075044, 2075045, 2075046.

### 3.4. DFT and TD-DFT Calculations

The ground state geometry optimization of the protonated *N*-oxides was computed through Gaussian 09 Rev. C.01 [52] software at DFT level of theory using the hybrid B3P86 functional and the triple-ζ *def2*-TZVP basis set [53]. In all cases, the polarizable continuum model (PCM) for water was used to take into account the effect of the solvent [54]. Harmonic frequency calculations were computed at the same level of theory for the ground state compounds which represented true minima on the potential energy surface (PES).

The electronic transitions were calculated at TD-DFT level of theory using PCM for water. Ten functionals were tested which are as follows: functionals based on the generalized gradient approximations (GGA) including CAM-B3LYP [55], ω-B97x-D [56], HSE06 [57], BH&HLYP [52]; the meta GGA functionals TPSSh [58], M06 [59], M06-2X [59], tHCTH [60] and the pure BLYP [61] and TPSS functionals [58]. All of them were combined with the *def2*-TZVP basis set. The molecular orbitals involved in the transitions were simulated through Mulliken population analysis at TPSSh/*def2*-TZVP level of theory. The representation of the UV-Vis spectra was generated using the GaussSum software [62].

## 4. Conclusions

The oxidation of substituted phen derivatives by PMS yields only the corresponding mono-*N*-oxides under acidic conditions because the intramolecular hydrogen bond involving the un-oxidized N and the N–O moiety inhibits further oxidation of the primary product. This resistance towards di-*N*-oxidation was utilized for the synthesis of a series of phenO-s. The reaction conditions are mild, the procedure is simple and results in the mono-*N*-oxides with good to excellent yields. The very nature of these compounds is still to be explored. They may prove valuable as specific ligands in coordination chemistry, as new precursors of building blocks in material science, materials for altering surfaces in electrochemical processes, etc. Our ongoing studies have already been directed toward evaluating these aspects of the chemistry of mono-*N*-oxides of 1-10 phenanthroline.

## Figures and Tables

**Figure 1 molecules-26-03632-f001:**
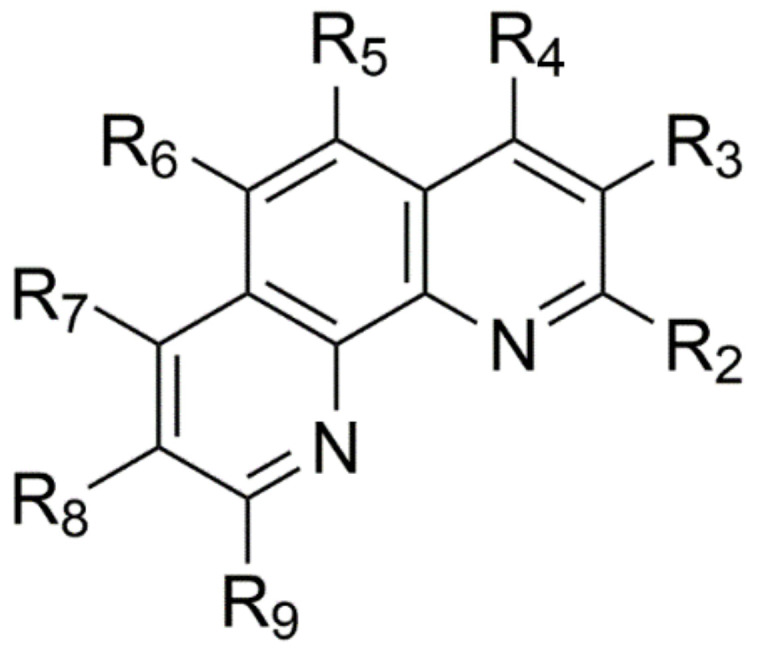
The general structural formula of the substrate. The substituents of the individual derivatives are listed in Table 1.

**Figure 2 molecules-26-03632-f002:**
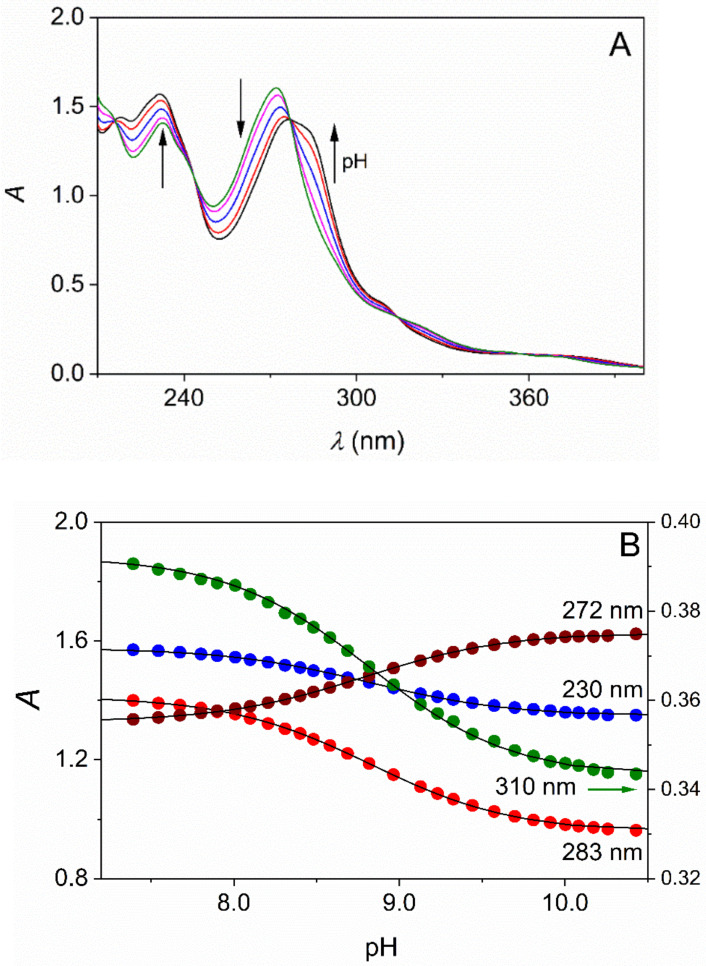
Representative pH-dependent spectra (**A**) and the absorbance at selected wavelengths (**B**) as a function of pH in aqueous solution of DMPO. [DMPO]tot = 0.45 mM, I = 1.00 M (NaCl), T = 25.0 °C, path length 1.000 cm. The arrows indicate the absorbance change upon the increase of the pH (**A**). The results of the simultaneous fit of the absorbance at the selected wavelengths to Equation (2) are shown as solid curves (**B**).

**Figure 3 molecules-26-03632-f003:**
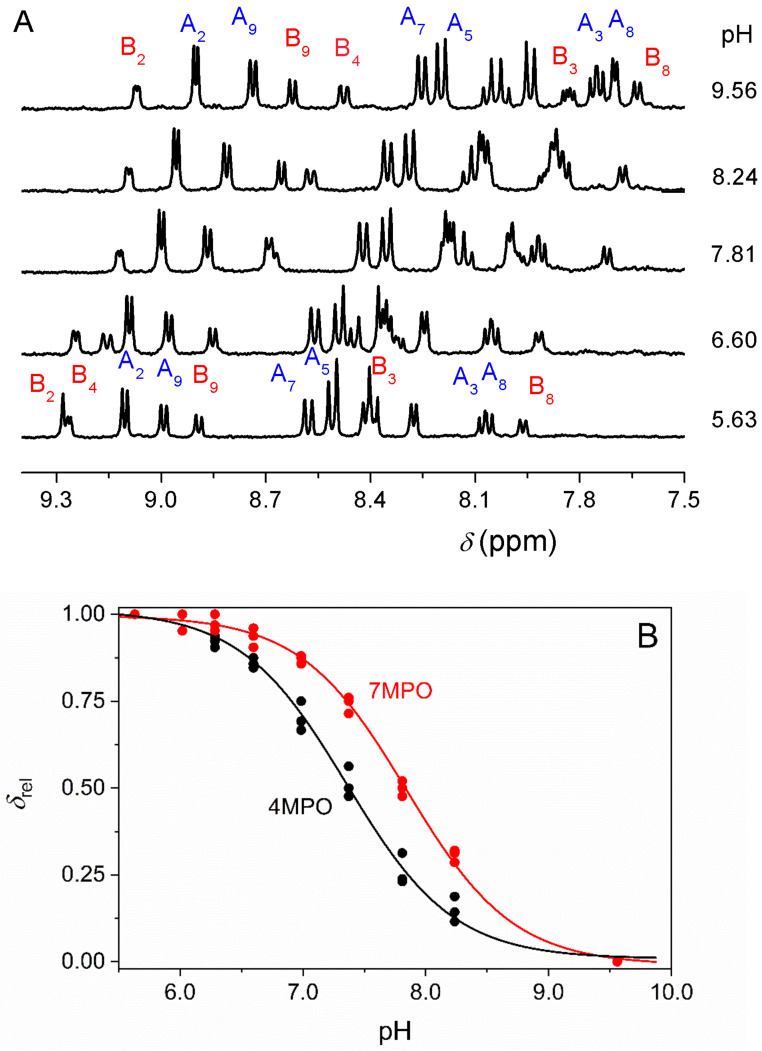
The pH dependence of the aromatic region of the ^1^H-NMR spectra of 4MPO (**A**) and 7MPO (**B**) in H_2_O (Figure 3A) and the plot of the relative chemical shift, *δ*_rel_, of selected peaks of 4MPO (methyl, A2, A9) and 7MPO (methyl, B2, B9) (Figure 3B) as a function of pH. The numbering of the peaks corresponds to Figure 1. Only every second spectrum is shown for clarity. The solid lines (4MPO: black, 7MPO: red) are obtained by simultaneously fitting the corresponding chemical shifts to equation 4. [4MPO+7MPO]_tot_ = 8.2 mM, *I* = 0.10 M, *T* = 25.0 °C.

**Figure 4 molecules-26-03632-f004:**
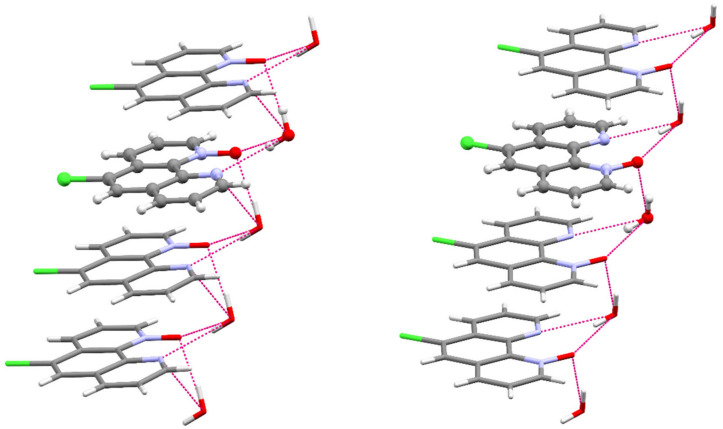
Partial packing view with selected strong hydrogen bonds of 5CPO×H_2_O (**left**) and 6CPO×H_2_O (**right**). The asymmetric unit is given in balls and sticks, and the symmetry-generated part of the compound is given in capped sticks representation.

**Figure 5 molecules-26-03632-f005:**
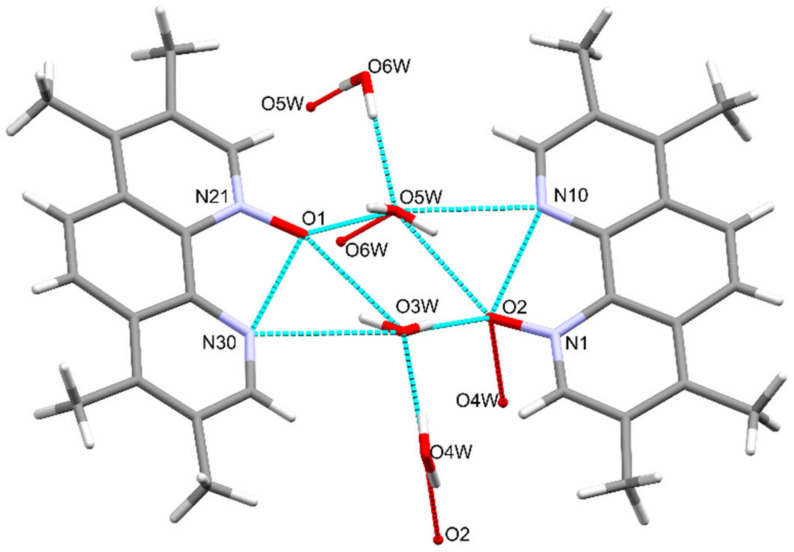
The hydrogen bond interactions between the TMPO and water molecules in 2×TMPO×4H_2_O.

**Figure 6 molecules-26-03632-f006:**
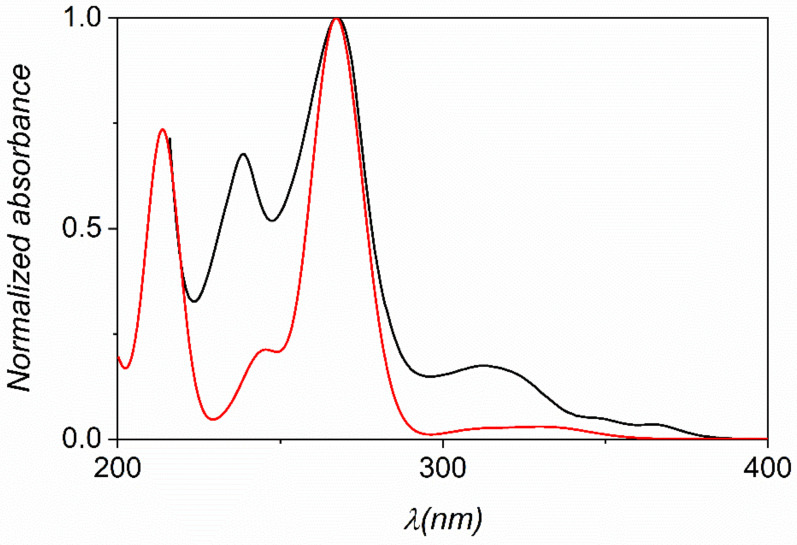
Experimental (**black**) and calculated (**red**) UV-Vis spectra of phenO using TD-DFT/TPSSh/*def2*-TZVP level of theory. The presence of water was taken into account by using the PCM approach.

**Figure 7 molecules-26-03632-f007:**
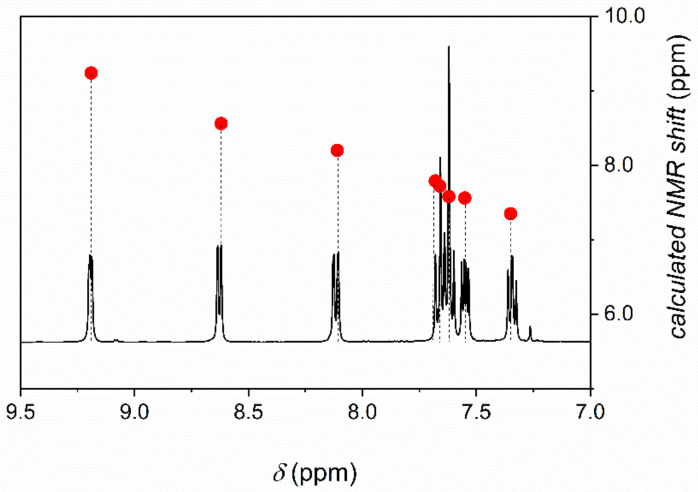
Experimental ^1^H NMR spectrum and calculated NMR shifts (right axis, ●) of phenO in chloroform. Calculations were performed at B3LYP/6-311+G(2d,p) level of theory. The effect of solvent was taken into account by adopting the PCM model for chloroform.

**Table 1 molecules-26-03632-t001:** The phen derivatives used in this study.

Name	Abbrev.	R_2_	R_3_	R_4_	R_5_	R_6_	R_7_	R_8_	R_9_
1,10-phenanthroline	phen	H	H	H	H	H	H	H	H
4-methyl-1,10-phenanthroline	4MP	H	H	Me	H	H	H	H	H
5-methyl-1,10-phenanthroline	5MP	H	H	H	Me	H	H	H	H
2,9-dimethyl-1,10-phenanthroline	DMP	Me	H	H	H	H	H	H	Me
3,4,7,8-tetramethyl-1,10-phenanthroline	TMP	H	Me	Me	H	H	Me	Me	H
5-nitro-1,10-phenanthroline	5NP	H	H	H	NO_2_	H	H	H	H
5-chloro-1,10-phenanthroline	5CP	H	H	H	Cl	H	H	H	H

**Table 2 molecules-26-03632-t002:** Synthesis of 1,10-phenanthroline-mono-*N*-oxides.

Product ^a^	Reaction Time (h)	Yield (%)
phenO	3.5	86.6
DMPO	10.0	91.2
TMPO	2.0	88.1
5MPO and 6MPO	3.5	79.5
5NPO and 6NPO	9.5	92.1
5CPO and 6CPO	38.0	87.2
4MPO and 7MPO	3.5	82.5

^a:^ phenO, DMPO, TMPO, 5MPO, 6MPO, 5NPO, 6NPO, 5CPO, 6CPO, 4MPO and 7MPO stand for the 1,10-phenanthroline-, 2,9-dimethyl-, 3,4,7,8-tetramethyl-, 5-methyl-, 6-methyl-, 5-nitro-, 6-nitro-, 5-chloro-, 6-chloro-, 4-methyl- and 7-methyl-1,10-phenanthroline-1-*N*-oxide, respectively.

**Table 3 molecules-26-03632-t003:** Acid dissociation constants of the phen derivatives.

Compound	p*K*_a_	Substrate	p*K*_a_
phen	5.12 ^a^		
phenO	7.10 ± 0.02 ^b^		
DMP	5.96 ^a^		
DMPO	8.83 ± 0.02 ^c^		
TMP	6.53 ^a^		
TMPO	9.03 ± 0.03 ^b^		
5MP	5.28 ^a^		
5MPO	7.40 ± 0.02 ^d^	6MPO	7.50 ± 0.03 ^d^
4MP	5.75 ^a^		
4MPO	7.36 ± 0.03 ^d^	7MPO	7.85 ± 0.02 ^d^
5CP	4.51 ^a^		
5CPO	6.40 ± 0.02 ^d^	6CPO	6.14 ± 0.03 ^d^
5NP	3.46 ^a^		
5NPO	5.41 ± 0.04 ^d^	6NPO	5.27 ± 0.03 ^d^

^a:^ Data are taken from Ref. [37]; ^b:^ by standard pH-potentiometric titration, *T* = 25.0 °C, *I* = 1.00 M (NaNO_3_); ^c:^ by combined pH-potentiometric and UV-vis titration, *T* = 25.0 °C, *I* = 1.00 M (NaCl).^d:^ by ^1^H NMR titration, *T* = 25.0 °C, *I* = 0.10 M (NaCl).

**Table 4 molecules-26-03632-t004:** The most important bond lengths (Å) of 5CPO×H_2_O, 6CPO×H_2_O and 6CPO.

	5CPO×H_2_O	6CPO×H_2_O	6CPO	JAJBAX ^a^	AZASEB ^b^
N1-O1	1.297(4)	1.306(2)	1.284(3)	1.284	1.320
C-Cl	1.738(4)	1.745(2)	1.740(2)	–	–
O1-O2W	2.870(4)	2.863(2)	–	2.747	2.819
N-O2W	3.012(5)	2.979(3)	–	3.479	2.836
plate-plate	3.494	3.392		6.667	3.358

^a^ 2,9-dimethyl-1,10-phenanthroline-1-*N*-oxide. Data are taken from Ref. [38]. ^b^ 4,10-phenanthroline-4-*N*-oxide. Data are taken from Ref. [41].

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
