# Peer review of "Synthesis and Characterization of 1,10-Phenanthroline-mono-N-oxides"

_molecules, 2021, doi:10.3390/molecules26123632_

Round 1

Reviewer 1 Report

this a good quality paper which can be published after minor revision. 

"N-oxides of heteroaromatic compounds" change to "N-oxides of nitrogen-containing heteroaromatic compounds" intext, abstract. 

Scheme 1 is not a Scheme it is Figure; make it a Scheme showing reaction; reagents and products.

Are these molecules perfectly flat? discuss the dihedral angles between C=N and C=N bonds. Why asymmetric crystallographic unit?

Reviewer 2 Report

The submitted study is of a high quality. It combines the results of synthesis, characterization and crystal structures studies of a series of compounds. The aim of this study is well motivated and the results are convincing.

Unfortunately, in the supplementary materials I have found only pdf files presenting checkcifs. I could not have found the rest of supplementary information files.

The introduction is very informative, concise and well written. The only thing that is missing is the explanation why these particular compounds have been chosen. Don’t get me wrong, I think that this choice is correct but were those compounds (listed in the Table 1) chosen for any particular reason?

Lines 77-86, I fully accept that by adjusting the pH you can control the type of final products. But have you also considered using another solvents for this purpose? Or a mixture of solvents?

Lines 115-117, since the Authors have used the DFT calculations in this study, they could have gone one step further and apply this molecular modelling method to explain those experimental findings by calculating the reaction pathway.

Figure 2A, please increase the size of this image. Besides, again, since the Authors were already using the DFT to support the UV-VIS analysis, why haven’t you calculated NMR properties using GIAO and compare the experimental and theoretical NMR results? It would prove your assumptions in a nice way.

Line 216, why only those three?

It would be nice if the Authors present the Figures comparing the crystal structure of 6CPO monohydrate and dehydrated forms. Besides, do you think it is possible to obtain the anhydrous 6CPO by the dehydration of its monohydrate? The water molecule seems to be very important in the stabilization of the structure of 6CPO monohydrate.

Round 2

Reviewer 2 Report

The Authors have answered my questions and I am satisfied with the answers.